# Real World Evaluation of the Prosigna/PAM50 Test in a Node-Negative Postmenopausal Swedish Population: A Multicenter Study

**DOI:** 10.3390/cancers14112615

**Published:** 2022-05-25

**Authors:** Una Kjällquist, Balazs Acs, Sara Margolin, Emelie Karlsson, Luisa Edman Kessler, Scarlett Garcia Hernandez, Maria Ekholm, Christine Lundgren, Erik Olsson, Henrik Lindman, Theodoros Foukakis, Alexios Matikas, Johan Hartman

**Affiliations:** 1Department of Oncology-Pathology, Karolinska Institute, 17164 Stockholm, Sweden; balazs.acs@ki.se (B.A.); emelie.karlsson.2@ki.se (E.K.); luisa.kessler@ki.se (L.E.K.); theodoros.foukakis@ki.se (T.F.); alexios.matikas@ki.se (A.M.); johan.hartman@ki.se (J.H.); 2Breast Center, Theme Cancer, Karolinska University Hospital, Solna, 17167 Stockholm, Sweden; 3Department of Clinical Pathology and Cancer Diagnostics, Karolinska University Hospital, 17176 Stockholm, Sweden; 4Department of Clinical Science and Education, Södersjukhuset, Karolinska Institute, 11883 Stockholm, Sweden; sara.margolin@regionstockholm.se; 5Department of Oncology, Södersjukhuset, 11828 Stockholm, Sweden; 6Breast Center, Capio St:Göran’s Hospital, 11235 Stockholm, Sweden; scarlett.garcia-hernandez@capiostgoran.se; 7Department of Oncology, Jönköping County, 55185 Jönköping, Sweden; maria.ekholm@rjl.se (M.E.); christine.lundgren@med.lu.se (C.L.); 8Department of Laboratory Medicine, Institute of Biomedicine, Sahlgrenska Center for Cancer Research, Sahlgrenska Academy at University of Gothenburg, 41345 Gothenburg, Sweden; 9Department of Oncology and Pathology, Institute of Clinical Sciences, Lund University, 22184 Lund, Sweden; 10Department of Immunology, Genetics and Pathology, Uppsala University, 75185 Uppsala, Sweden; erik.olsson@regionuppsala.se (E.O.); henrik.lindman@igp.uu.se (H.L.)

**Keywords:** adjuvant chemotherapy, breast cancer, gene expression signature, gene expression profiling, impact, Prosigna, PAM50

## Abstract

**Simple Summary:**

Gene expression signatures can provide important information on the risk of recurrence in patients with hormone receptor positive early breast cancer, and they can guide postoperative treatment. We have investigated how the implementation of gene-expression-based risk signatures with the Prosigna^®^ test impacted patient management in Sweden. The two major conclusions of this study are that prognostic factors derived from routine pathology were poor predictors of the intrinsic subtype and the risk of recurrence score, and that gene-expression-based risk combined with clinicopathological biomarkers (tumor size, Ki67, tumor grade) spared patients from adjuvant chemotherapy, but also identified patients who would potentially benefit from this treatment.

**Abstract:**

Molecular signatures to guide decisions for adjuvant chemotherapy are recommended in early ER-positive, HER2-negative breast cancer. The objective of this study was to assess what impact gene-expression-based risk testing has had following its recommendation by Swedish national guidelines. Postmenopausal women with ER-positive, HER2-negative and node negative breast cancer at intermediate clinical risk and eligible for chemotherapy were identified retrospectively from five Swedish hospitals. Tumor characteristics, results from Prosigna^®^ test and final treatment decision were available for all patients. Treatment recommendations were compared with the last version of regional guidelines before the introduction of routine risk signature testing. Among the 360 included patients, 41% (*n* = 148) had a change in decision for adjuvant treatment based on Prosigna^®^ test result. Out of the patients with clinical indication for adjuvant chemotherapy, 52% (*n* = 118) could avoid treatment based on results from Prosigna^®^ test. On the contrary, 23% (*n* = 30) of the patients with no indication were escalated to receive adjuvant chemotherapy after testing. Ki67 could not distinguish between the Prosigna^®^ risk groups or intrinsic subtypes and did not significantly differ between patients in which decision for adjuvant therapy was changed based on the test results. In conclusion, we report the first real-world data from implementation of gene-expression-based risk assessment in a Swedish context, which may facilitate the optimization of future versions of the national guidelines.

## 1. Introduction

Adjuvant chemotherapy (ACh) reduces the risk of recurrence and breast cancer (BC)-related death up to one third regardless of any clinical or pathologic factor, such as estrogen receptor (ER) expression [1]. Importantly, for postmenopausal women with node-negative ER-positive and Human Epidermal Growth Factor Receptor 2 (HER2)-negative disease, the derived absolute benefit is less pronounced due to the lower absolute risk for disease recurrence. Thus, the selection of patients for ACh, especially in the intermediate risk group, should be balanced between known prognostic factors (for example, tumor size, grade and proliferation) and acute and late toxicities and, importantly, comorbidity. 

With the significant advances in the understanding of the underlying complexity and heterogeneity of BC biology [2,3,4], gene expression profiles (GEP) to select patients with low risk of recurrence have been developed, validated and regulatory approved [5]. Several retrospective studies using real-world data have demonstrated a marked decline in adjuvant chemotherapy use, especially in patients with node-negative disease [6,7,8,9]. Clinical utility has been shown prospectively for two signatures (Recurrence Score and 70-gene assay), with a marked reduction up to 20–35% in chemotherapy usage and no negative effect in long-term survival for postmenopausal patients with ER-positive, HER2-negative BC and low to intermediate risk of gene-expression-based recurrence [10,11,12]. Several of these signatures, including the 70-gene assay/MammaPrint (Agendia Inc., Amsterdam, The Netherlands), Oncotype Dx recurrence score (RS) (Exact Sciences Corp., Madison, WI, USA), EndoPredict/EPclin (EP, Myriad Genetics Inc., Salt Lake City, UT, USA), and Prosigna^®^/risk of recurrence (ROR) (Veracyte Inc., South San Francisco, CA, USA) are recommended for clinical use by the American Society of Clinical Oncology (ASCO) and the European Society for Medical Oncology (ESMO) [7,13,14,15].

Based on health technology assessment, OncotypeDx^®^ and Prosigna^®^ are both recommended for clinical use in Sweden [16]. Prosigna^®^ is an assay that determines the intrinsic subtype and risk of recurrence (ROR) score based on the 50-gene Predictor Analysis of Microarray 50 (PAM50) gene signature and tumor size [17]. ROR assigns patients into three risk groups (low, intermediate and high) to predict the long-term risk of distant recurrence in both node-negative and node-positive patients, faring favorably in the limited direct comparisons with other prognostic tools [17,18,19,20]. Risk signature-guided adjuvant therapy for specific age and anatomical-stage-based subgroups of ER-positive/HER2-negative BC has been recommended by the Swedish national breast cancer guidelines since 2019, but the clinical implementation has been relatively conservative due to concerns regarding risk for undertreatment [21]. In addition, the cost–benefit of Prosigna^®^ testing has been judged as uncertain due to lack of national data to estimate the rate of ACh administered in the intended population. As a result, the rate of treatment was assumed to increase due to implementation of Prosigna^®^ testing in the health economic analysis by the The Dental and Pharmaceutical Benefits Agency (Tandvårds- och läkemedelsförmånsverket, TLV) [16]. Furthermore, concerns were raised by the Swedish agency for medical technology testing (Medicintekniska produktrådet, MTP) about the added value of Prosigna^®^ in comparison with risk assessment based on routine diagnostics, including Ki67, due to lack of such data [22]. The objective of this study was to assess the real-life impact of implementing routine Prosigna^®^ testing in a Swedish multicenter context. The study clarifies some of the concerns and presents a detailed analysis on the impact of treatment based on clinical and biological tumor characteristics. 

## 2. Materials and Methods

### 2.1. Population

Patients tested with Prosigna^®^ according to regional guidelines during the study period, March 2020 to March 2022, were included from five different hospitals in Sweden (Karolinska University Hospital, Södersjukhuset and St:Görans hospital in Stockholm, Akademiska Hospital in Uppsala, and Ryhov Hospital in Jönköping), which cover three of Sweden’s twenty-one regions comprising a population of approximately three million (approximately 30% of the Swedish population). Data were collected retrospectively from electronic patient charts to a predefined form, and included patient and tumor characteristics, Prosigna^®^ test results and administered treatment. All participating centers follow the national guidelines of biomarker evaluation and participate in internal and external quality control of passing breast cancer biomarker testing [23]. According to regional guidelines for gene expression risk signatures to guide clinical decision making during the study period, Prosigna^®^ was recommended for postmenopausal women with node-negative, ER-positive and HER2-negative BC, at an intermediate clinical risk of recurrence. The classification of risk of recurrence was based on pathologic characteristics, including tumor size, grade and Ki67. These guidelines are presented in detail in Table 1. 

### 2.2. Assessment of Clinical Impact of Prosigna Testing

Each patient was classified regarding the indication for ACh according to prior regional guidelines from 2018, before gene expression risk testing was introduced as a recommendation in the Swedish national breast cancer guidelines, here referred to as 2018 Stockholm guidelines and presented in Table 2. The impact on given treatment based on Prosigna^®^ was determined as de-escalation (ACh omitted despite prior clinical indication), escalation (ACh administered despite no prior clinical indication), or not affected by testing.

### 2.3. Biomarker Assessment

According to the Swedish guidelines [23], estrogen receptor (ER) and progesterone receptor (PR) status are considered positive when ≥10% of tumor cells show ER- and PR-specific staining in tumor nuclei detected by immunohistochemistry (IHC). ER status (Clone SP1) was considered as a binary variable, either positive (≥10%) or negative (<10%). HER2 status was considered either positive or negative, detected by IHC (Clone 4B5) and/or ISH (Roche/Ventana SISH Inform Her-2 DUAL ISH) as recommended by the Swedish guidelines; IHC was performed first, followed by amplification testing in case of a 2 + IHC score. The Swedish guidelines are following the ASCO/CAP guidelines in reporting HER2 status [24]. PR status (Clone 1E2 or Clone 16) was considered as a continuous variable and defined as the percentage of tumor cells with positive nuclear staining evaluated globally on a section of the representative tumor block. Ki67 (Clone 30-9 or Clone MIB-1) proliferation index was analyzed as a continuous variable and defined as the percentage of tumor cells with positive nuclear staining counted in a hot-spot with a minimum of 200 tumor cells (in accordance with last version of Swedish guidelines from 2020). All the IHC analysis was run using Ventana Benchmark Ultra (Roche). Histological grading was performed on hematoxylin and eosin (HE)-stained slides according to the Nottingham grade scoring system [25].

### 2.4. PREDICT 2.1 

PREDICT is an online tool that provides at an individual patient level prognostic information according to clinical, pathological and treatment variables [26]. The latest update is version 2.1 [27]. Results for the NHS PREDICT tool were obtained by the R package nhs.predict version 1.4.0, using R software (version 4.1.2, Vienna, Austria, http:// cran.r-project.org, accessed on 26 March 2022).

### 2.5. Statistical Analysis

Normal distribution was tested by Kolmogorov–Smirnov test of normality, and nonparametric methods were used for significance testing. Kruskal–Wallis test with Dunn’s post-hoc analysis, Mann–Whitney test, Spearman correlation with Kendall rank correlation coefficient, and χ2 tests were used to compare the variables with one another. We applied Bonferroni correction in multiple comparisons. To interpret the results, scatterplots with linear regression, boxplots, bar charts and pie charts were used. In all statistical analyses, the level of significance was set at *p* < 0.05. For statistical analysis, SPSS 25 software was used (IBM, Armonk, New York, USA).

## 3. Results

### 3.1. Population

A total of 360 postmenopausal women with ER-positive, HER2-negative and node negative BC were included from five Swedish hospitals. The patients’ clinical and demographic characteristics are shown in Table 3. Prosigna^®^ classified 58.3% of tested individuals (*n* = 210) as molecularly Luminal A, 40.3% (*n* = 145) as Luminal B and a small fraction (0.8%) as HER2-enriched. No patients had basal-like tumors. 

### 3.2. Clinicopathologic Correlates

Due to the strong prognostic value of the proliferation marker Ki67 in early ER-positive BC [28] and the high dependence of ROR score on proliferation [29], we first investigated the relationship between Prosigna^®^ and Ki67. Ki67 showed moderate correlation to ROR score (Spearman’s rho = 0.56; Figure 1a) across ROR subgroups. In addition, Ki67 values differed significantly between ROR subgroups (low, intermediate and high risk; Mann–Whitney *p*< 0.001 for all comparisons; Figure 1b) and between intrinsic subtypes (Luminal A and Luminal B; Mann–Whitney *p* = 0.001; Appendix A). However, Ki67 values showed weak correlation with ROR (Spearman’s rho < 0.3; Appendix A) in the intrinsic subtype groups and clearly overlapped between ROR subgroups. Furthermore, no significant difference was observed in Ki67 expression between the patient groups in which treatment was changed after Prosigna^®^ testing (*p* = 0.954) (Figure 2d). 

PR expression is used in the St Gallen surrogate definitions to distinguish between Luminal A-like and Luminal B-like tumors [28], which forms the basis of the prior 2018 Stockholm guidelines. However, even though PR is a positive prognostic factor [30], data do not support PR levels as a predictor for ACh benefit [1]. The subtype predictive role has also been questioned [31]. Here, PR expression showed no significant correlation to ROR risk groups or luminal subgroups (Mann–Whitney *p* = 0.69), or between ROR risk groups (Mann–Whitney *p* >0.05) (Appendix A).

Histologic grade was associated with both ROR subgroups (Pearson’s chi-square *p*< 0.001, Mann–Whitney *p* <0.05) as well as intrinsic subtypes (Pearson’s chi-square *p*< 0.001). ROR and Ki67 across tumor grades are presented in Figure 3 and Appendix A. 

### 3.3. Impact on Clinical Decision Making

ROR score and treatment recommendation from the multidisciplinary tumor board were available for all 360 patients included in the study. According to the 2018 Stockholm guidelines, 225 patients (62.5%) had an indication for ACh, with 210 (58%) and 145 (40%) being classified according to the surrogate definitions as Luminal A and Luminal B, respectively. 

In total, 148 patients (41.7%) had a change in the decision for ACh based on Prosigna^®^ test results. More specifically, 118 patients (32.8%) were de-escalated (ACh omission); for 103 patients (28.6%), the decision to not administer ACh was confirmed; for 107 patients (29.7%), the decision to administer ACh was confirmed; and finally, 30 patients (8.3%) that would not have been recommended ACh were escalated following testing with Prosigna^®^ (Figure 2a,b). Out of the 225 patients with indication for ACh, 118 (52.0%) were de-escalated, and out of the 133 patients with no indication for ACh according to 2018 Stockholm guidelines, 30 (22.6%) were escalated based on Prosigna^®^ test result. As a result, 137 patients actually received ACh following testing, representing a net decrease in the use of chemotherapy of 39.1%. In this postmenopausal, node-negative population, 3.1 Prosigna^®^ tests were needed to avoid ACh for one patient. This was more prominent for grade 2 tumors (2.7 tests per avoided chemotherapy), in tumors with high Ki67 (2.5 tests per avoided chemotherapy) and in tumors >20 mm (2.8 test per avoided chemotherapy). These data are summarized in Table 4. 

Ki67 differed significantly between the two groups in which Prosigna^®^ did not affect treatment recommendations (ACh versus no ACh, Mann–Whitney *p* = 0.001; Figure 2d). In contrast, Ki67 did not differ between patients that had a change in treatment decision based on Prosigna^®^ ROR, i.e., escalated versus de-escalated (Mann–Whitney *p* = 0.954). This underscores the potential added value of gene-expression-risk-based classification to routine markers. Furthermore, out of 12 patients with grade 1 tumors, only one had a change in treatment decision based on testing. Because grade 2 tumors are a large and ambiguous group in terms of risk for recurrence, risk stratification by routine markers is less reliable. Not surprisingly, most patients whose treatment was changed based on Prosigna^®^ had grade 2 tumors (128 patients, 86.5%). Out of these, 106 patients (82.8%) could omit treatment, while 22 (17.2%) were recommended for ACh (Figure 2c). Interestingly, even patients with histologically grade 3 tumors had treatment change following Prosigna^®^ (22 patients, 14.9%), with 13 patients (59.1%) omitted and 9 patients (40.9%) receiving ACh. For these grade 3 tumors, size was significantly smaller (Mann–Whitney *p* = 0.0002; average size 8.3 mm versus 16.2 mm) in the de-escalated group, implying an underestimation of small grade 3 tumors in prior guidelines, whereas there was no difference in terms of Ki67 (Mann–Whitney *p* = 0.45).

Finally, we assessed whether ACh benefit estimated by the NHS PREDICT tool could forecast the change in treatment decision after Prosigna^®^ test. The PREDICT model predicts adjuvant treatment benefit and prognosis based on the combination of clinical and pathological characteristics and can aid in decision making without the need for time consuming and costly gene expression signatures [27]. Although not approved in regulation, PREDICT online tool is recommended for use in clinical practice according to UK National Institute for Health and Clinical Excellence (NICE) guidelines 2018 as a complement to treatment guidelines, especially for postmenopausal women with ER-positive, HER2-negative, node-negative breast cancer [32]. Therefore, we wanted to investigate if PREDICT could assess treatment guidance in line with ROR. No difference in expected treatment benefit by PREDICT was observed when comparing patients with ACh indication (without treatment change versus ACh omitted, Mann–Whitney *p* = 0.30) or between patients without ACh indication (no ACh versus escalation, Mann–Whitney *p* = 0.10; Appendix A). In addition, the correlation between expected treatment benefit by PREDICT and ROR (Spearman’s rho = 0.18) and Ki-67 (Spearman’s rho = 0.13) was poor (Appendix A). Thus, there was no correlation between the estimated benefit of ACh from PREDICT and the ROR score.

## 4. Discussion

Evaluation of gene expression risk signatures in clinical practice is important to better appreciate both the clinical utility in terms of reduced chemotherapy use without risk of undertreatment, as well as the cost–benefit, and adjustment of guidelines for test indication. So far, there are limited real-life data evaluating Prosigna^®^ [33,34], but in a French study on 809 women with ER-positive, HER2-negative tumors, a 44% reduction in chemotherapy use and an estimated positive cost–benefit were reported [35]. Similarly, an evaluation of EndoPredict in a German population showed that 38% had a change in treatment decision and 25% withheld chemotherapy [36]. 

The implementation of risk signatures to guide adjuvant chemotherapy decisions in BC has been relatively slow in Sweden, and although recommended in the national guidelines, the health economic benefit has been considered unclear due to uncertainties of the fraction of patients with indication for testing and concerns about increased use of chemotherapy. Here we report the first evaluation of risk signatures in clinical practice from five Swedish hospitals. We observed a 52% decrease in the number of indications for chemotherapy based on Prosigna^®^ test results. Out of these, 82% were grade 2 with highly variable tumor characteristics. This both confirms the utility of gene expression profiling with Prosigna^®^ and underscores that testing should be performed for all patients with ambiguous tumor characteristics and who are fit for chemotherapy. Of the patients with no indication for chemotherapy according to regional guidelines, 32 patients (23%) were escalated to receive ACh based on Prosigna^®^ test results, nearly all (29 patients) with a Luminal B subtype despite small tumor size < 20 mm (27 patients) and histological grade 2 (23 patients). This highlights the importance of testing to avoid undertreatment also among tumors with an estimated modest risk from clinical routine markers. Importantly, PR expression showed no association to ROR or PAM50 subtype, questioning thus its recommendation for distinguishing Luminal A and Luminal B subtypes. Furthermore, Ki67 overlapped substantially between ROR risk groups and was not significantly different between groups in which patients had their treatment decision changed based on ROR testing, i.e., escalation and de-escalation, underscoring the fact that Ki67 is not a suitable marker to predict intrinsic subtypes [37] and indicating that ROR score is required to further discriminate ambiguous cases with intermediate to high Ki67 expression. This uncertainty was true also for NHS PREDICT estimations, and we observed no agreement between PREDICT adjuvant treatment benefit estimates and ROR score. These two tools would therefore likely not result in the same treatment decision, which is an important observation for clinical practice. These results underscore the ambiguity of intermediate clinical risk group and the importance of implementing risk signature testing to refine prognostication. Based on the regional guidelines applied before introduction of the Prosigna^®^ test, the fraction of candidates for ACh was 63% for all tested patients and 43% for the subgroup (grade 2 and size >10 mm) for which the health economic analysis was based [16], exceeding thus the previously anticipated impact. 

In Sweden, candidates for gene expression testing are selected based on tumor size, tumor grade, Ki67 and menopausal status. Most tumors with indication for testing in this study were grade 2 and intermediate to high proliferation score. Although Ki67 showed a modest association with ROR, Ki67 is subject to inter-observer variability resulting in low analytical performance [38], which highlights the role of Prosigna^®^ in Ki67 intermediate/NHG2 patients. On the other hand, Ki67 below 6% or above 29%, in addition to NHG1 or NHG3, respectively, are robust prognostic thresholds that, despite residual analytical variability even after standardization, are sufficient for clinical decision making on the need for adjuvant chemotherapy [39].

Limitations in this study that should be acknowledged include the lack of detailed cost–benefit estimations, which are ongoing. Furthermore, no follow up is available yet, although it is likely that the results from long-term studies with similar demographics would be comparable [9]. Although the study population is limited, patients are represented from five different hospitals in three Swedish regions, providing a relevant scope. Finally, selection bias is likely considering the predetermined guidelines for whom to test and the variable adherence to those guidelines, depending on factors not captured by our study, such as clinical decision that a patient would not tolerate ACh regardless of her clinical or molecular risk for recurrence. Moreover, this study has specifically looked at the clinical consequences of the Prosigna^®^ test and cannot be automatically extrapolated to other gene expression risk signatures that may result in other recommendations.

We conclude that Prosigna^®^, when combined with clinicopathological biomarkers (tumor size, Ki67, NHG), adds important clinical utility and improved risk stratification, reduces the use of chemotherapy, but also identifies high-risk ER positive/HER2 negative, N0, postmenopausal patients who would not receive treatment based on only routine clinicopathology.

## Figures and Tables

**Figure 1 cancers-14-02615-f001:**
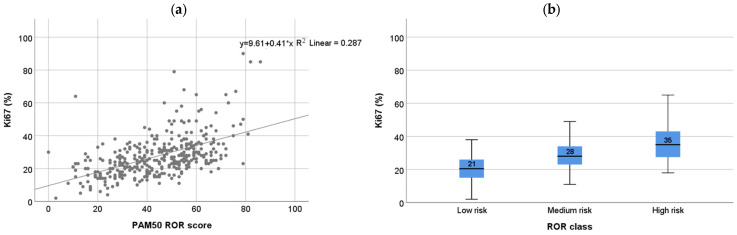
Correlation between Ki67 and ROR (**a**) Ki67% versus Prosigna risk of recurrence (ROR) score; Spearman correlation: rho = 0.555 (*p* < 0.001, *n* = 360); (**b**) Ki67% in ROR risk class (low 0–40, medium, 41–60, high 61–100); Mann–Whitney *p* < 0.001 between all groups.

**Figure 2 cancers-14-02615-f002:**
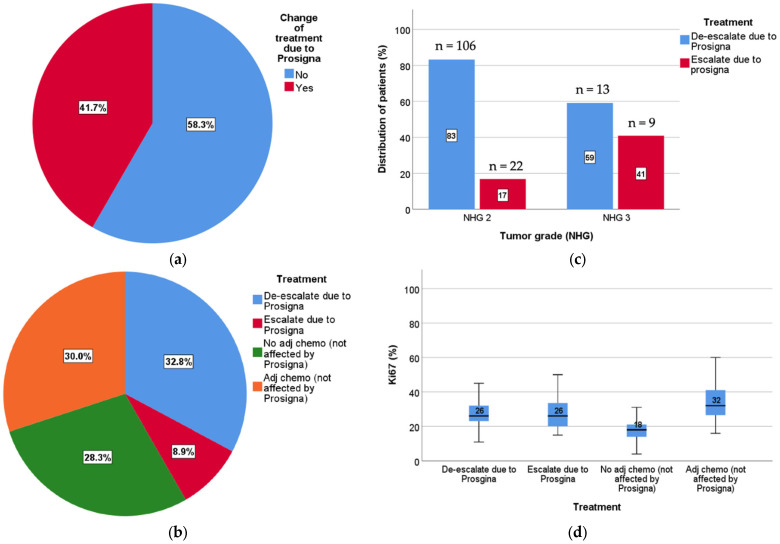
Impact on treatment decision after Prosigna test. (**a**) Percentage of patients in which treatment decision was changed based on ROR results (blue) or confirmed (red); (**b**) the fraction of patients who could avoid ACh (blue) or needed to receive ACh (red) based on ROR results. Patients not affected by ROR testing either were treated (orange) or not treated (green) with ACh based on routine tumor characteristics and treatment guidelines; (**c**) change of treatment due to Prosigna in NHG groups. Of all patients impacted by testing, 83% of grade 2 tumors were de-escalated (blue) and 17% escalated (red). Among grade 3 tumors, 59% were de-escalated (blue) and 41% escalated (red); (**d**) change of treatment due to Prosigna vs. Ki67. No difference in Ki67 expression was seen between the patients with a treatment change (de-escalated versus escalated) (*p* = 0.954). Patients not affected by test results showed different Ki67 expression in the ACh versus non-ACh groups (*p* = 0.001).

**Figure 3 cancers-14-02615-f003:**
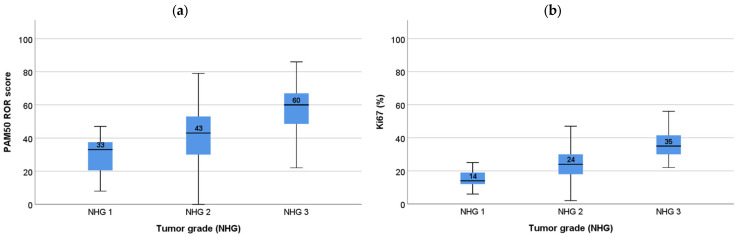
ROR and Ki67 in tumor grade; (**a**) Risk of recurrence score (ROR) versus Nottingham Histologic Grade (NHG). Mann–Whitney NHG 1 vs. NHG 2 *p* = 0.031; NHG 1 vs. NHG 3 *p* < 0.001; NHG 2 vs. NHG 3 *p* < 0.001; (**b**) Ki67 (%) versus NHG. Mann–Whitney *p* < 0.001 between all groups.

**Table 1 cancers-14-02615-t001:** Indication for Prosigna testing according to Swedish national guidelines from 2019 and adapted regionally. NHG; Nottingham Histologic Grade. * Pathology lab specific Ki67 limits; Karolinska, Södersjukhuset, St:Göran hospital low 0–14%, intermediate 15–22%, high 23–100%; Jönköping, low 0–11%, intermediate 12–21%, high 22–100%; Akademiska hospital, low 0–14%, intermediate 15–23%, high 24–100%.

2019 Stockholm Guidelines for Adjuvant Chemotherapy or Prosigna Test in ER-Positive (ER ≥ 10%), HER2-Negative, Node Negative, Postmenopausal Patients			
Tumor Size	Low-Risk LumA-Like NHG-1, Low Ki67 * and PR ≥ 20%	Intermediate-Risk LumA/B-Like NHG-1/2 Intermediate Ki67 *	High-Risk LumB-Like NHG-3 or NHG-1/2 and High Ki67 *
<5 mm	No Chemo	No Chemo	No Chemo
6–10 mm	No Chemo	No Chemo	Prosigna
11–20 mm	No Chemo	Prosigna	Prosigna
>20 ≤ 50 mm	Prosigna	Prosigna	Prosigna or Chemo
>50 mm	Prosigna	Chemo	Chemo

**Table 2 cancers-14-02615-t002:** Indication for adjuvant chemotherapy according to Swedish national guidelines from 2018 and adapted regionally. NHG; Nottingham Histologic Grade. * Pathology lab specific Ki67 limits; Karolinska, Södersjukhuset, St:Göran hospital low 0–14%, intermediate 15–22%, high 23–100%; Jönköping, low 0–11%, intermediate 12–21%, high 22–100%; Akademiska hospital, low 0–14%, intermediate 15–23%, high 24–100%.

2018 Stockholm Guidelines for Adjuvant Chemotherapy or Prosigna Test in ER-Positive, HER2-Negative, Node Negative Patients > 35 Years		
Tumor Size	LumA-Like ER ≥ 50%, NHG-1/2, Low to Intermediate Ki67 * and PR ≥ 20%	LumB-Like ER ≥ 10% and NHG-3 or NHG-2 and High Ki67 * or Intermediate Ki67 and PR < 20%
<5 mm	No Chemo	No Chemo
6–10 mm	No Chemo	No Chemo
11–20 mm	No Chemo	Chemo
>20 ≤ 50 mm	No Chemo	Chemo
>50 mm	Chemo	Chemo

**Table 3 cancers-14-02615-t003:** Tumor and patient characteristics. N; lymph node status, NHG; Nottingham Histologic Grade, PR; progesterone receptor, ROR; risk of recurrence score. * Ki67 pathology lab specific limits are presented in Table 1.

	n/Mean	%/Range
Age average (years)	65.1	(41–84)
Tumor size (mm)	19.4	(6–60)
N0	358	99.4%
N+	2	0.6%
Micrometastasis	4	1.1%
Tumor grade		
NHG 1	12	3%
NHG 2	285	79%
NHG 3	62	17%
Ki67 (average)	27.7%	(2–90)
Low *	27	7.5%
Intermediate *	99	27.5%
High *	234	65.0%
PgR	58.2%	0–100
≥20%	274	76%
<20%	86	24%
PAM50 Prosigna		
LumA	210	58.3%
LumB	145	40.3%
HER2-enriched	3	0.8%
ROR score	44	0–86
low risk (ROR 0–40)	146	40.6%
intermediate risk (ROR 41–60)	153	42.5%
high risk (ROR 60–100)	56	15.6%

**Table 4 cancers-14-02615-t004:** Number of tests required per omitted adjuvant chemotherapy per tumor characteristic (tumor grade, proliferation Ki67and T tumor size) and additional benefit of chemotherapy (%) on 10-year recurrence-free survival (rfs) based on NHS Predict score. * Ki67 pathology lab specific limits are presented in Table 1.

	Number of Patients Omitted Chemo	Number of ProSigna Tests	ROR Risk Category			Number of Tests Needed to Avoid One Chemotherapy Treatment
			High	Intermediate	Low	
All patients	118	360	56	153	146	3.1
Grade						
2	105	285	29	127	129	2.7
3	13	63	29	26	-	4.8
Ki67						
Low *	3	27	0	3	29	9.0
Intermediate *	23	99	3	33	57	4.3
High *	93	234	56	118	61	2.5
T1 ≤ 20mm	77	244	34	111	99	3.2
T2 > 20mm	40	113	25	43	48	2.8
Predict chemo benefit (10 year rfs)						
0–3%	80	283	33	123	127	3.5
3–6%	33	64	19	27	18	1.9
6–9%	4	13	7	4	2	3.3
Grade 2, T >10 mm	105	256	26	112	117	2.4
Grade 2, T >10 mm, Low Ki67	5	26	-	3	23	5.2
Grade 2, T >10 mm, Intermediate Ki67	19	86	3	32	51	4.5
Grade 2, T >10 mm, High Ki67	81	144	24	77	43	1.8

## Data Availability

The data presented in this study are available on request from the corresponding author.

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
