# Peer review of "Real World Evaluation of the Prosigna/PAM50 Test in a Node-Negative Postmenopausal Swedish Population: A Multicenter Study"

_cancers, 2022, doi:10.3390/cancers14112615_

Round 1
Reviewer 1 Report
The manuscript written by Una Kjallquist et.al investigates how implementation of Prosigna/PAM50 based gene expression has impacted the patient management in node negative post-menopausal Swedish population. The data has been evaluated from 5 Swedish hospitals. The manuscript is very well written. The manuscript organization is easy to follow and well structured. The language is simple and easy to understand.
There are few minor typos,
Please see the below revisions:
- Page 7 of 13, line 226-240 is also repeated on page 8 of 13, line 256-270. Please revise.
Author Response
Dear reviewer,
thank you for taking the time to review our work and for your positive comments. The manuscript has been revised accordingly.
Reviewer 2 Report
The evaluation in routine clinical practice of the impact of genomic assays on the decision of adjuvant chemotherapy in intermediate risk HR positive/Her2 negative breast cancer has been addressed by several retrospective and prospective studies conducted with the various genomic assays available showing on average a reduction in chemotherapy prescription of 30-40%
A number of studies have also attempted to determine whether the standard routinely available clinical and pathological features were associated with the genomic assay risk categories and could be considered a reliable surrogate of their results.
As in other studies with different multigene assays, proliferation measured by KI67 was the feature more strictly associated with genomic assay results despite the inconsistent data on the definition of a predictive cut-off for ki67.
The results point out the added value of the Prosigna results over the classical clinical and pathological features especially in the intermediate risk group.
The results of the present study are consistent with previous findings of other real-world series in terms of increased prognostic accuracy as compared with clinicopathological features and of likelihood of a decrease of the indication of adjuvant chemotherapy.
The formatting of table 1a is not clear
I advise to specify in the text the cut-off used to classify Ki67 in low, intermediate and high subgroups rather than in the supplementary material
The authors should specify the rationale for investigating a correlation between the Prosigna results and the Predict tool which evaluates grossly classified clinical and pathologic characteristics of the patients (age, menopausal status) and the tumor (ER positive or negative, ki 67 high or low ) in addition to the previous correlations with detailed pathological characteristics and which results they expected from this analysis
The meaning of the last sentence of the Conclusions should be clarified.
Author Response
Dear reviewer,
thank you for taking the time to review our work.
Point 1: The formatting of table 1a is not clear. I advise to specify in the text the cut-off used to classify Ki67 in low, intermediate and high subgroups rather than in the supplementary material
Response 1: Table 1 has been updated and clarified. The Ki67 cut-off ranges has been added to the figure text line 126-129: * Pathology lab specific Ki67 limits; Karolinska, Södersjukhuset, St:Göran hospital low 0–14%, intermediate 15-22%, high 23-100%; Jönköping, low 0–11%, intermediate 12-21%, high 22-100%; Akademiska hospital, low 0–14%, intermediate 15-23%, high 24-100%.
Point 2: The authors should specify the rationale for investigating a correlation between the Prosigna results and the Predict tool which evaluates grossly classified clinical and pathologic characteristics of the patients (age, menopausal status) and the tumor (ER positive or negative, ki 67 high or low ) in addition to the previous correlations with detailed pathological characteristics and which results they expected from this analysis.
Response 2:
Very valid comment. We added this analysis in order to see if using the predict tool could forecast the treatment decision in the same direction as ROR. As predict is used frequently in the clinic and even recommended by NICE 2018 before molecular testing it would be important to see if there is an agreement.
we´ve added this rationale according to your suggestion.
Line 277-294:
Finally, we assessed whether ACh benefit estimated by the NHS PREDICT tool could forecast the change in treatment decision after Prosigna® test. The PREDICT model predicts adjuvant treatment benefit and prognosis based on the combination of clinical and pathological characteristics can aid in decision making without the need for time consuming and costly gene expression signatures [27]. Although not regulatory approved, PREDICT online-tool is recommended for use in clinical practice according to UK National Institute for Health and Clinical Excellence (NICE) guidelines 2018 as a complement to treatment guidelines, especially for postmenopausal women with ER-positive, HER2-negative, node-negative breast cancer [32]. Therefore, we wanted to investigate if PREDICT could assess treatment guidance in line with ROR. No difference in expected treatment benefit by PREDICT was observed when comparing patients with ACh indication (without treatment change versus ACh omitted, Mann-Wh hitney p = 0.30) or between patients without ACh indication (no ACh versus escalation, Mann-Whitney p = 0.10; Supplementary Figure S 4a and b). In addition, the correlation between expected treatment benefit by PREDICT and ROR (Spearman’s rho = 0.18) and Ki-67 (Spearman’s rho = 0.13) was poor (Supplementary Figure S4c and d). Thus, there was no correlation between the estimated benefit of ACh from PREDICT and the ROR score.
Line 327-330 (Discussion):
This uncertainty was true also for NHS PREDICT estimations and we observed no correlation between PREDICT adjuvant treatment benefit estimates and ROR score. These two tools will therefore likely not result in the same treatment decision, which is a clinically important observation that could be further investigated.
Point 3:
The meaning of the last sentence of the Conclusions should be clarified.
Response 3:
We´ve clarified the implications of the poor correlation between predict and ROR, se prior comment. Last sentence in discussion removed, was a formatting error from the word template.
Reviewer 3 Report
I love reading this manuscript. It uses non-interventional collection and analysis of data from patient records to explore the question if Prosigna when combined with tumor size, Ki67 and NHG facilitate the prediction of Adjuvant therapy in treating BC. Statistical method is correct and the analyses are good.
A couple of minor revision suggestion,
"hormone receptor (HR)" is context dependent and not specific. Please specify "estrogen receptor (ER)" or "progesterone receptor (PR)". According to section 2.3 HR and discussion lines 204-210. "HR" in the abstract and the body of this manuscript is more likely referring to ER. Authors, please address this clearly by replacing HR with ER wherever it is appropriate.
"Ki-67" or "Ki67" should be consistent throughout the manuscript. Please unify the term usage.
Author Response
Dear reviewer,
thank you for taking the time to review our work and for your positive response!
Point 1: "hormone receptor (HR)" is context dependent and not specific. Please specify "estrogen receptor (ER)" or "progesterone receptor (PR)". According to section 2.3 HR and discussion lines 204-210. "HR" in the abstract and the body of this manuscript is more likely referring to ER. Authors, please address this clearly by replacing HR with ER wherever it is appropriate.
Response 1: Thank you for pointing this out. We´ve revised as suggested.
Point 2:
"Ki-67" or "Ki67" should be consistent throughout the manuscript. Please unify the term usage.
Response 2: Thank you, we´ve adressed this.
Reviewer 4 Report
This is an interesting study which reflect the real world data about gen expression profile in breast cancer patient. However, most important of gene expression study is that they did improve clinical outcome. Thus I would like to know in those who had change clinical management, the had equal or better out come compare with their counter part? (for example, for high clinical risk patient omit adjuvant chemotherapy had the same outcome as those who had recieved adjuvant chemotherapy.)
Author Response
Dear reviewer,
thank you for taking the time to review our work and for your positive response!
Response:
Thank you for you important comment. This study was designed as a retrospective collection of patient data to investigate how the clinical use of Prosigna testing affected the treatment decision in clinical practice. While survival data and follow up would be of great importance we do not have access to this data and therefore we could not provide this analysis in this manuscript.
Round 2
Reviewer 4 Report
This study shown imporotant real world data, i would recommend for publication